# Hospital Admission in the Latent versus the Active Phase of Labor: Comparison of Perinatal Outcomes

**DOI:** 10.3390/children9060924

**Published:** 2022-06-20

**Authors:** Viola Seravalli, Noemi Strambi, Enrica Castellana, Maria Alessia Salamina, Chiara Bettini, Mariarosaria Di Tommaso

**Affiliations:** Department of Health Sciences, Division of Obstetrics and Gynecology, Careggi University Hospital, University of Florence, Largo Brambilla, 50134 Florence, Italy; viola.seravalli@unifi.it (V.S.); enrica.castellana@stud.unifi.it (E.C.); alessia.salamina@gmail.com (M.A.S.); bettini.chiara@yahoo.it (C.B.); mariarosaria.ditommaso@unifi.it (M.D.T.)

**Keywords:** labor, phase of labor, epidural analgesia, caesarean section, obstetrics intervention

## Abstract

Background: Admission in the latent phase of labor has been associated with increased risk of obstetric interventions compared to admission in the active phase. We aimed to investigate the relationship between labor phase at admission and obstetric and neonatal outcomes. Methods: A retrospective cohort study was conducted on 1005 women with uncomplicated singleton pregnancy admitted for spontaneous labor. Cesarean section rate and other perinatal outcomes were compared between women admitted in the latent phase and those admitted in the active phase. Results: Admission occurred in the active phase of labor for 331 women (32.9%) and in the latent phase for 674 (67.1%). Admission in the latent phase was more frequent in nulliparous than in multiparous (*p* < 0.01) and for Italian patients compared to foreigners. The incidence of caesarean section was similar between groups. Admission in the latent phase increased the likelihood of epidural analgesia (OR 3.47, 95% CI 1.96–6.14, in nulliparous, and OR 2.58, 95% CI 1.37–4.84, in multiparous) and increased the rate of augmentation of labor with oxytocin in multiparous (OR 2.87, 95% CI 1.05–7.85), without difference in neonatal outcomes. Conclusions: Admission in the latent phase is associated with more frequent use of epidural analgesia, without an increase in cesarean section or adverse neonatal outcomes.

## 1. Introduction

For pregnant women with labor onset at home, especially if nulliparous, it may be very difficult to determine the right time to go to the hospital. Women who come to the hospital in early labor are often admitted when they are still in the latent phase. Postponing admission until the onset of the active phase of labor is a suggested approach to reduce obstetric interventions in women with spontaneous labor at term, with the fetus in vertex presentation. This decision should be individualized according to maternal and fetal risks and status [1]. However, while some studies found an association between admission in the latent phase of labor and an increased risk of obstetric interventions, prolonged hospital stays, and higher health care costs [2,3,4,5,6,7,8,9,10,11,12], others did not [13,14]. For instance, the use of epidural analgesia and oxytocin for augmentation of labor were found to be three and two times more frequent, respectively, in women admitted in the latent phase than in those admitted in the active phase of labor [6,7]. On the other hand, whether admission in the latent phase of labor increases the risk of cesarean section is still a matter of debate, as different studies have reported conflicting results [4,5,11,13,14]. Currently, the World Health Organization (WHO) recommends delaying labor ward admission until the active first stage only in research settings [15].

Unlike other countries, in Italy the national healthcare system does not offer the possibility of home visits by midwifes in early labor, a service that may contribute to better labor management, although it does not seem to have any clear effect on maternal and neonatal outcomes [16].

The aim of this study was to examine the relationship between labor phase on admission and perinatal outcomes in a tertiary care center in Italy. 

## 2. Materials and Methods

This retrospective cohort study was conducted between January 2017 and October 2018 at Careggi University Hospital in Florence, Italy. Women with uncomplicated singleton pregnancy at term (≥37 weeks), who were admitted to the hospital in either the active or the passive phase of spontaneous labor, and who subsequently delivered, were included. Women with premature rupture of membranes at admission and those with high-risk pregnancy (eg. cases of maternal diabetes or hypertension, fetal malformations, abnormal fetal growth) were excluded. The final sample comprised 1005 women (Figure 1). 

All data were collected from electronic medical charts. The cervical dilatation at the time of admission was identified for each woman and was used to distinguish two groups: the latent phase of labor group, if the cervical dilatation was <4 cm, and the active phase of labor group, if the cervical dilatation was ≥4 cm. In accordance with the protocol in use at our hospital, women admitted in the latent phase were sent to the obstetric inpatient ward and later transferred to the delivery room upon diagnosis of the active phase of labor, while women admitted in the active phase were sent directly to the delivery room. Demographic and obstetric characteristics, such as maternal age, nationality, body mass index (BMI), parity, and mode of conception were recorded. 

The primary endpoint was to compare the incidence of cesarean section (CS) between women admitted in the latent phase and those admitted in the active phase of labor, after dividing nulliparous from multiparous women. The secondary endpoint was the comparison of other obstetric and neonatal outcomes between groups, including the use of epidural analgesia, labor augmentation with oxytocin, major post-partum hemorrhage (defined as a blood loss > 1000 cc), length of hospital stay, birthweight, cord pH, 1 min Apgar score < 7, need for resuscitation, and admission to neonatal intensive care unit (NICU).

Statistical analysis was performed using a Graph Pad INSTAT3 software package (San Diego, CA, USA). The Shapiro-Wilk test was used to evaluate the distribution of data, and showed that all continuous variables had a non-parametric distribution (*p* < 0.05). We used the non-parametric Mann-Whitney test to compare the continuous variables and the Chi-square test to compare the categorical variables between the two groups (latent vs. active phase of labor at admission). The odds ratios (OR) and 95% confidence intervals (CI) for the associations of the labor phase at admission with each outcome were also calculated.

An “a priori” analysis was carried out to determine the sample size using G*Power software (vers. 3.1.7). The calculation was based on the values reported in the literature for the incidence of CS in women admitted in latent labor (CS incidence 21.8%) or in active labor (CS incidence 14.5%) [4]. Based on these values, using a significance level of 0.05 and a power level of 0.80, with a 2:1 ratio between groups, the calculated sample size was 1002 patients in total (668 women admitted in latent labor and 334 women admitted in active labor).

The study was conducted according to the guidelines of the Declaration of Helsinki, and approved by the local Ethics Committee (protocol code 17631, 30 June 2020).

## 3. Results

Of the 1005 women included in the study, 331 (32.9%) were admitted in active labor and 674 (67.1%) were admitted in latent labor. Patients’ characteristics are reported in Table 1. There were no significant differences between groups for maternal age, BMI and mode of conception. Multiparous women, and those of foreigner origin, were more likely to be admitted in the active phase than in the latent phase of labor (*p* < 0.01 and *p* = 0.02, respectively).

Results of the comparison between the latent phase and the active phase admission groups are depicted in Table 2. Among both nulliparous and multiparous women, CS occurred with similar incidence between the latent and the active phase groups (10.4% vs. 7.8%, *p* = 0.57, in nulliparous, and 1.0% vs. 0.9%, *p* =1.00, in multiparous). Admission in the latent phase of labor was associated with increased likelihood of epidural analgesia when compared to admission in the active phase (OR 3.47, 95% CI 1.96–6.14 in nulliparous, and OR 2.58, 95% CI 1.37–4.84, in multiparous women). Among multiparous women, admission in latent labor was also associated with more frequent need for labor augmentation with oxytocin (OR 2.87, 95% CI 1.05–7.85).

No differences were observed in the length of maternal hospital stay nor in neonatal outcomes between the latent-phase and the active-phase admission groups, in nulliparous or multiparous women (Table 2).

## 4. Discussion

Our results show that nulliparous women were admitted more frequently in the latent phase of labor, while multiparity was associated with fewer early hospital admissions. These findings are in accordance with previous studies [2,3,4,5,6,7,8,9,10,11]. We also observed that, compared to Italian women, foreigner women were more likely to be admitted in active labor. Similarly, Iobst et al. reported a higher incidence of admission at dilation of 0–3 cm in White non-Hispanic women than in women of other ethnicities [9].

Considering our primary outcome, the lack of a significant difference in the incidence of cesarean rate in women admitted in the latent phase compared to those admitted in the active phase of labor is in agreement with the results of two previous studies on a similar cohort of low-risk nulliparous women with spontaneous labor at term [13,14]. In contrast to our findings, some previous studies have reported an association between early admission and increased likelihood of cesarean section [2,3,4,5,6,7,8,9,10,11,12]. However, some of these studies do not specify if the pregnancies included were low- or high-risk [4,9], or do not distinguish between nulliparous and multiparous [6]. Kauffman et al. found a higher rate of cesarean delivery in women admitted in the latent phase of labor [4], but they included women with obesity, preeclampsia, or pregestational diabetes, which are all risk factors for CS [17,18,19,20]. According to Iobst et al., CS was five times more frequent in women admitted with cervical dilatation of 0–3 cm than in women admitted with a dilatation > 6 cm [9]. However, it is uncertain if admission in the latent phase of labor itself raises the risk for CS or whether women requiring admission earlier in labor are more likely to have an anomalous labor course [21]. While unnecessary admissions and interventions should be avoided, there is insufficient evidence to directly relate admission in the latent phase to increased risk of cesarean section. Moreover, the lack of agreement on the definition of active labor makes it difficult to compare results between studies.

Admission in latent labor in our cohort was associated with a higher risk of epidural analgesia, in agreement with previous studies [2,4,6,10,13]. Another potential consequence of early hospital admission is a longer duration of fetal and maternal monitoring, and often medical interventions to accelerate labor [4,8,9,11,13]. Iobst et al. showed that women were twice as likely to receive a combination of epidural, oxytocin and amniotomy when admitted at ≤3 cm dilatation [9]. In our cohort, admission in latent labor was associated with increased likelihood of labor augmentation with oxytocin in multiparous women (6.0% vs. 2.2%, *p* = 0.03). This result is in agreement with previous studies, although the frequency of augmentation with oxytocin reported by other authors is higher than in our cohort (ranging from 20 to 46% in multiparous women admitted in the latent phase).

As expected, we did not find a statistically significant difference in neonatal outcomes between groups, in accordance with Holmes et al. [10], who analyzed Apgar score and cord pH as neonatal outcomes in a cohort of low-risk nulliparous and multiparous women. Kauffman et al. found an increase in NICU admissions in infants of multiparous women admitted in the latent phase, but this result was not observed among nulliparous women [4].

Strengths of this study include the large sample size, which reached the required number of patients to evaluate our primary outcome with sufficient power, and the selection of low-risk patients, which prevented the confounding effect that inclusion of high-risk pregnancies would have on the association between labor phase at admission and obstetric outcomes. Furthermore, the incidence of c-section in our cohort of low-risk nulliparous women was 10%, which is comparable to the rate of low-risk cesarean deliveries (12.7%) reported by the Society for Maternal-Fetal Medicine [22], and by Bailit et al. in a cohort of low-risk nulliparous women (10.1%) [11], thus supporting the external validity of our results.

The main limitation of our study is its retrospective design. Some information is lacking: in particular, we do not know if some women had prior access to the hospital and were sent home for “false labor”. Another limitation is that the physician assessing cervical dilation at admission is usually the one on duty in the triage room, and therefore it was not the same person for all patients. Variability in the assessment of cervical dilation is a relevant confounder in studies on labor progress.

## 5. Conclusions

Admission in the latent phase compared to the active phase of labor is associated with increased likelihood of epidural analgesia, without a significant increase in the risk of cesarean section or adverse neonatal outcome. Implementation of midwifery service and support interventions might contribute to decrease medical intervention during labor and to raise maternal satisfaction with care [16,23].

## Figures and Tables

**Figure 1 children-09-00924-f001:**
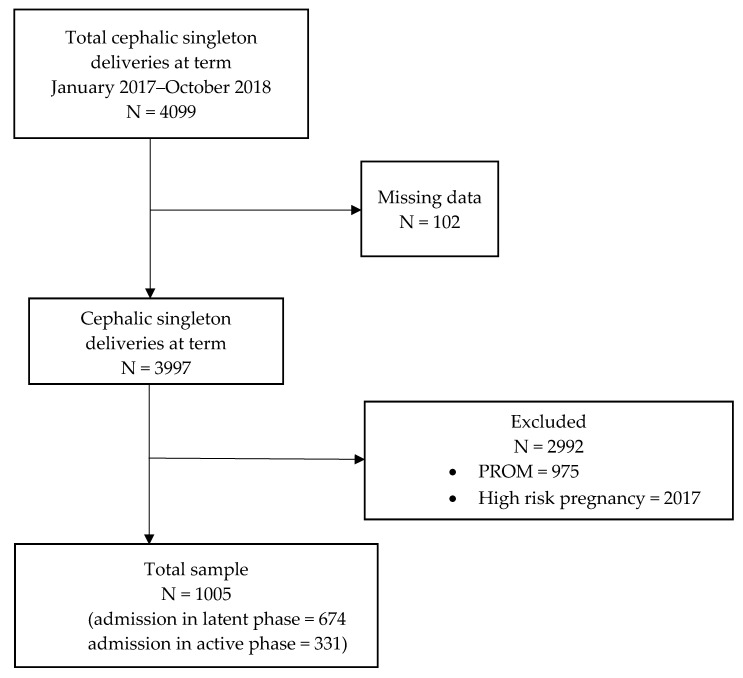
Patient enrollment flow chart. PROM: Premature rupture of membrane.

**Table 1 children-09-00924-t001:** Comparison of demographic and obstetric characteristics of low-risk women admitted in active or latent phase labor at term.

	Latent Phase (*n* = 674)	Active Phase (*n* =331)	*p*
Maternal Age (years)	30.87 ± 5.61	30.88 ± 5.46	0.98
BMI ≥ 25 kg/m^2^	74 (11.0%)	37 (11.2%)	0.92
Nationality			
Italian	464 (68.8%)	204 (61.6%)	0.02
Foreigner	210 (31.2%)	127 (38.4%)	
Parity			
Nulliparous	375 (55.6%)	102 (30.8%)	<0.01
Multiparous	299 (44.4%)	229 (69.2%)	
Assisted Reproductive Technologies	16 (2.4%)	6 (1.8%)	0.65
Gestational age at delivery	39.81 ± 1.78	39.71 ± 0.92	0.35

**Table 2 children-09-00924-t002:** Comparison of maternal and neonatal outcomes between women admitted in the latent-phase and those admitted in the active phase of labor. NICU, neonatal intensive care unit.

	Latent Phase (*n* = 674)	Active Phase (*n* =331)	*p*	Odds Ratio (95% CI)
**Nulliparous women**	**375**	**102**		
*Maternal and delivery outcomes*				
Epidural analgesia	147 (39.2%)	16 (15.7%)	<0.01	3.47 (1.96–6.14)
Augmentation of labor with oxytocin	54 (14.4%)	14 (13.7%)	1.00	1.06 (0.56–1.99)
Cesarean delivery	39 (10.4%)	8 (7.8%)	0.57	1.36 (0.61–3.02)
Vaginal delivery	336 (89.6%)	94 (92.2%)	0.57	0.73 (0.33–1.62)
Major post-partum hemorrhage	5 (1.3%)	1 (1.0%)	1.00	1.37 (0.16–11.82)
Hospital stays (days)	3.01 ± 1.03	2.86 ± 1.39	0.14	-
*Neonatal outcomes*				
Birthweight (g)	3300± 347	3319 ± 345	0.58	
Need for resuscitation	1 (0.3%)	0	1.00	0.82(0.03–20.32)
Cord pH	7.25 ± 0.09	7.27 ± 0.09	0.06	
Cord pH < 7.2	68 (18.1%)	12 (11.8%)	0.14	1.66 (0.86–3.21)
1 min Apgar score < 7	4 (1.1%)	0	0.58	2.48 (0.13–46.53)
NICU admission	20 (5.3%)	4 (3.9%)	0.80	1.38 (0.46–4.13)
**Multiparous women**	**299**	**229**		
*Maternal and delivery outcomes*				
Epidural analgesia	43 (14.4%)	14 (6.1%)	<0.01	2.58 (1.37–4.84)
Augmentation of labor with oxytocin	18 (6.0%)	5 (2.2%)	0.03	2.87 (1.05–7.85)
Cesarean delivery	3 (1.0%)	2 (0.9%)	1.00	1.15 (0.19–6.94)
Vaginal delivery	296 (99.0%)	227 (99.1%)	1.00	0.87 (0.14–5.25)
Major post-partum hemorrhage	6 (2.0%)	5 (2.2%)	1.00	0.91(0.28–3.04)
Hospital stays (days)	3.54 ± 1.41	3.27 ± 1.51	0.07	-
*Neonatal outcomes*				
Birthweight (g)	3392 ± 371	3425 ± 353	0.22	
Need for resuscitation	0	1 (0.4%)	0.43	0.25 (0.01–6.27)
Cord pH	7.29 ± 0.09	7.29 ± 0.08	0.94	
Cord pH < 7.2	37 (12.4%)	30 (13.1%)	0.90	0.94 (0.56–1.57)
1 min Apgar score < 7	2 (0.7%)	2 (0.9%)	1.00	0.76 (1.11–5.47)
NICU admission	13 (4.3%)	12 (5.2%)	0.68	0.82 (0.37–1.84)

## Data Availability

Data will be made available by the corresponding author upon reasonable request.

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
