# Peer review of "Hospital Admission in the Latent versus the Active Phase of Labor: Comparison of Perinatal Outcomes"

_children, 2022, doi:10.3390/children9060924_

Round 1

Reviewer 1 Report

Overall, this is an interesting paper on a clinically relevant issue. Authors should provide some more information regarding the protocol for admitting women in the labour ward. What are the criteria? Are all women admitted irrespective of presenting at the latent or the active phase?

Author Response

1) Overall, this is an interesting paper on a clinically relevant issue. Authors should provide some more information regarding the protocol for admitting women in the labour ward. What are the criteria? Are all women admitted irrespective of presenting at the latent or the active phase?We thank the reviewer for the remark and we have now included more details about the admission to the labor ward in the Methods section, as follows: “In accordance with the protocol in use at our hospital, women admitted in the latent phase were sent to the obstetric inpatient ward and later transferred to the delivery room upon diagnosis of active phase of labor, while women admitted in the active phase were sent directly to the delivery room.”

2) Moderate English changes required.  The manuscript has now been reviewed by a native English speaker to correct any grammatical or typos errors.

Reviewer 2 Report

Professional articles with sufficiently presentation of the aim, methods and results.

Accept after minor revision: english language

Author Response

1) Professional articles with sufficiently presentation of the aim, methods and results.Accept after minor revision: english language.  The manuscript has now been reviewed by a native English speaker to correct grammatical or typos errors.
2) Does the introduction provide sufficient background and include all relevant references? Must be improved. We thank the reviewer for the suggestion. We have now improved the introduction section.
